# Monitoring and Treatment of Paroxysmal Nocturnal Hemoglobinuria in Patients with Aplastic Anemia in Asia: An Expert Consensus

**DOI:** 10.3390/ijms252212160

**Published:** 2024-11-13

**Authors:** Raymond Siu Ming Wong, Jun Ho Jang, Lily Lee Lee Wong, Jin Seok Kim, Ponlapat Rojnuckarin, Yeow-Tee Goh, Yasutaka Ueda, Wen-Chien Chou, Jong Wook Lee, Yuzuru Kanakura, Tzeon-Jye Chiou

**Affiliations:** 1Sir Y.K. Pao Centre for Cancer & Department of Medicine and Therapeutics, Prince of Wales Hospital, The Chinese University of Hong Kong, 30-32 Ngan Shing Street, Sha Tin, Hong Kong SAR, China; 2Division of Hematology-Oncology, Department of Medicine, Samsung Medical Center, Sungkyunkwan University School of Medicine, 2066 Seobu-Ro, Suwon 16419, Republic of Korea; 3Queen Elizabeth Hospital, 13a, Jalan Penampang, Kota Kinabalu 88200, Sabah, Malaysia; 4Division of Hematology, Department of Internal Medicine, Yonsei University College of Medicine, Severance Hospital, 50-1 Yonsei-ro, Sinchon-dong, Seodaemun-gu, Seoul 03722, Republic of Korea; 5Department of Medicine, Faculty of Medicine, Chulalongkorn University and King Chulalongkorn Memorial Hospital, 1873 Rama IV Rd, Pathum Wan, Bangkok 10330, Thailand; rojnuckarinp@gmail.com; 6Research Unit in Translational Hematology, Chulalongkorn University, 254 Phaya Thai Rd, Khwaeng Wang Mai, Pathum Wan, Bangkok 10330, Thailand; 7Department of Haematology, Singapore General Hospital, Outram Rd, Singapore 169608, Singapore; 8Department of Hematology and Oncology, Graduate School of Medicine, Faculty of Medicine, Osaka University, 1-1 Yamadaoka, Suita, Osaka 565-0871, Japan; yueda@bldon.med.osaka-u.ac.jp; 9Division of Hematology, Department of Internal Medicine, National Taiwan University Hospital, No.7, Chung Shan S. Rd. (Zhongshan S. Rd), Zhongzheng Dist., Taipei City 100225, Taiwan; 10Division of Hematology-Oncology, Hanyang University Seoul Hospital, 222-1 Wangsimni-ro, Seong-dong-gu, Seoul 04763, Republic of Korea; 11Department of Hematology, Sumitomo Hospital, 5-chōme-3-20 Nakanoshima, Kita Ward, Osaka 530-0005, Japan; 12Cancer Center, Division of Hematology and Oncology, Department of Medicine, Taipei Municipal Wanfang Hospital, Taipei Medical University, No. 111, Section 3, Xinglong Rd, Wenshan District, Taipei City 11696, Taiwan; 108178@w.tmu.edu.tw

**Keywords:** aplastic anemia, Asia–Pacific, clone, flow cytometry, paroxysmal nocturnal hemoglobinuria, treatment

## Abstract

Paroxysmal nocturnal hemoglobinuria (PNH) clones can be identified in a significant proportion of patients with aplastic anemia (AA). Screening for PNH clones at the time of an AA diagnosis is recommended by national and international guidelines. In this report, an expert panel of physicians discusses current best practices and provides recommendations for managing PNH in patients with AA in the Asia–Pacific region. Plasma/serum lactate dehydrogenase (LDH) levels and reticulocyte count should be measured with every blood test. PNH clone size should be monitored regularly by flow cytometry, with on-demand testing in the event of a rise in LDH level ± reticulocyte count or development of symptoms such as thrombosis. Monitoring for PNH clones can guide the choice of initial AA treatment, although flow cytometry has resource implications which may present a challenge in some Asia–Pacific countries. The treatment of patients with both PNH and AA depends on which condition predominates; following PNH treatment guidelines if hemolysis is the main symptom and AA treatment guidelines if bone marrow failure is severe (regardless of whether hemolysis is mild or moderate). The expert panel’s recommendations on the monitoring and treatment of PNH in patients with AA are practical for healthcare systems in the Asia–Pacific region.

## 1. Introduction

Paroxysmal nocturnal hemoglobinuria (PNH) and aplastic anemia (AA) are rare disorders, with reported incidences of 1–3.5 cases/million people/year [1,2] and 2–7.4 cases/million/year [3], respectively. Both diseases are more common in Asian than in Western populations [1,3]. PNH clones can be identified in a significant proportion of patients with AA [4,5,6]. As PNH disease status may impact the overall management plan for a patient with AA, early diagnosis is important. Various guidelines about AA and PNH have been published, although many do not address monitoring for PNH clones in detail [7,8,9,10,11,12,13,14,15,16]. Determining PNH clone size by flow cytometry has resource implications which could be challenging in some Asia–Pacific countries. In addition, little information is available to determine the optimal treatment strategy for patients who have both AA and PNH.

To address these issues, an expert panel comprising the authors of this article from Hong Kong (China), Japan, Malaysia, the Republic of Korea, Singapore, Taiwan and Thailand convened to discuss best practices and provide consensus recommendations on the monitoring and management of PNH in patients with AA in the Asia–Pacific region. Panel members experienced in the treatment of patients with PNH and AA were drawn from departments of Medicine, Hematology and Hematology–Oncology. In preparation for the meeting, a search of PubMed was conducted to identify the relevant literature on monitoring and treatment of patients with PNH and AA, including national/international guidance. During the meeting, the panel reviewed the available literature, discussed current practices in their countries, and addressed the following questions: (a) How do you screen and monitor for PNH among AA patients? What indicators should be monitored and how frequently? Should clone size be monitored regularly? What cutoff should be used for a clinically meaningful PNH clone size? (b) What criteria drive AA treatment plan changes in patients with PNH? What should be the overall treatment and monitoring plan for patients with PNH and AA? Which guidelines, if any, do you follow in your practice? Based on the outcome of the discussion, the panel made recommendations for PNH monitoring and management in Asian patients with AA.

Herein, we present an overview of AA and PNH, the main tests used to monitor PNH clones in patients with AA, and treatment options, together with recommendations for the monitoring and management of AA patients with PNH in the Asia–Pacific region.

## 2. Background

This section provides an overview of AA and PNH, the association between the two disorders, and the relevance of PNH clone size in AA patients.

### 2.1. Aplastic Anemia: General Considerations

AA is an archetypal bone marrow failure disease that is classified as inherited or acquired. Inherited forms of AA are rare genetic diseases that are well characterized clinically and associated with identifiable germline mutations [17]. Of the rare and diverse group of heterogeneous diseases associated with bone marrow failure, idiopathic acquired AA (aAA) is the most common with a reported incidence of 2–14 cases per million per year; the higher rates are reported in Asia [17]. aAA is an autoimmune-related, non-malignant disorder characterized by a loss of bone marrow function, hypocellularity of the bone marrow and peripheral cytopenias [18]. Symptoms of aAA are typical of a peripheral blood cytopenia and can include fatigue, pale skin, severe bruising and tachycardia (as a result of anemia), increased risk of infections (as a result of low white blood cell count), and increased risk of bleeding, bruising, reduced clotting ability and petechiae (as a result of a low platelet count) [19]. Over the years, numerous etiologies for aAA have been proposed. Recent research suggests multimodal involvement of both the adaptive and innate immune systems, culminating in the autoreactive cytotoxic T-cell destruction of hematopoietic stem cells and progenitors which contribute to T-cell aberrant clonal expansion [20].

### 2.2. Aplastic Anemia Clonal Hematopoiesis

Historically regarded as a relatively benign condition devoid of clonal abnormalities, it has become increasingly recognized that somatic alterations (chromosomal abnormalities and single nucleotide variants) contribute to the pathogenesis of aAA [21]. The introduction of next-generation sequencing has identified a number of somatic mutations linked to bone marrow failure and, in combination with other genomic sequencing techniques, is providing a better understanding of the molecular mechanisms and genetic changes involved [20,21]. The most frequent somatic mutations that occur in aAA involve the *BCOR/BCOR1* and *PIGA* (phosphatidylinositol glycan-A) genes and the loss of human leukocyte antigen alleles (6pLOH) [17,20,22]. Mutations involving PIGA lead to a reduction in or absence of glycosylphosphatidylinositol (GPI) expression on cell membranes, and a subsequent absence of GPI-bound proteins [23].

Clonal hematopoiesis, i.e., an over-representation of blood cells from a single clone, occurs in healthy individuals as well as in patients with benign or malignant diseases (e.g., myeloid neoplasia) and has been linked to the aging process [20,22]. It is also common in patients with aAA. Clonal evolution from aAA to secondary PNH or myeloid neoplasia represents a long-term complication in such patients.

### 2.3. Aplastic Anemia and Paroxysmal Nocturnal Hemoglobinuria

The overlap between aAA and PNH has been well documented for more than 50 years [24]. Across seven clinical studies, the presence of PNH clones was observed in 25–62% (median 40%) of patients with AA [4,7,25,26,27,28,29]. Similar results (21–69% AA patients with a PNH clone) were reported in Asian countries including Japan [4,5,30], Thailand [31] and China [32]. Rates of PNH positivity below <25% were recorded in patients with severe or refractory AA [5,32]. PNH clones can vary in size from very small/small to large. Many have a transient lifespan with levels of the remainder fluctuating over time [4,31]. Sugimori and colleagues investigated levels of PNH-type cells over a period of 5 years in 75 patients with bone marrow failure (61% with AA), and found that the level increased in 17%, persisted in 59% and disappeared in 24% [4]. The presence of PNH clones (irrespective of size) was shown to be associated with a more favorable response to immunosuppressive therapy in Asian patients [4,5,30,32], consistent with findings from other studies and a meta-analysis [6,33]. Interestingly, the presence of PNH-type cells also predicted a good response to treatment with eltrombopag [5]. The results of the meta-analysis suggest that a positive PNH clone at the time of AA diagnosis is associated with an increased risk of developing PNH/AA-PNH syndrome after immunosuppressive therapy (odds ratio 2.78, *p* = 0.016) [33].

PNH is a severe, life-threatening disorder for which early diagnosis is essential. Clonal expansion of hematopoietic cells deficient in GPI-linked protein renders them susceptible to complement-induced destruction. Furthermore, clonal expansion of *PIGA*-mutated stem cells results in either complete or partial absence of CD55 (decay accelerating factor) and CD59 (membrane inhibitor of reactive lysis), two GPI-dependent molecules that are complement activity regulators. Red blood cells lacking these proteins are sensitive to complement-mediated intravascular hemolysis. The unrestricted action of complement at the surface of hematopoietic cells, including platelets and leukocytes, initiates a complex chain of pathophysiological events that culminates in an increased risk of thrombosis [34]. The root cause of these changes is somatic mutations in hematopoietic stem cells, leading to the complete or partial absence of complement-regulatory proteins on blood cells and subsequent intravascular hemolysis, thrombosis and bone marrow failure [1,35]. In terms of clonal complications, the development of myeloid neoplasia in patients with PNH/aAA is a risk and generally has a poor prognosis. It was calculated to occur in about 12% of PNH/aAA patients over 10 years [36].

Given the high incidence of PNH clones in patients with aAA, and the associated additional risk it confers, it is important to identify their presence early and monitor patients, since the development of clinical PNH may exacerbate morbidity and mortality. In addition, PNH status may impact the overall treatment and management plan for patients who also have AA.

### 2.4. Aplastic Anemia and Paroxysmal Nocturnal Hemoglobinuria Clone Size

Flow cytometry studies have shown a bimodal size distribution of granulocyte PNH clones which correlates closely with clinical presentation. The majority (two-thirds) of clone cell populations are small (granulocyte clones usually <30%; mean clone size ~11%) and occur in patients with bone marrow failure/cytopenia (mostly aAA). PNH symptoms and clinical hemolysis are absent, and the risk of thrombosis is low. However, one-third of patients have features of classical PNH with large clone size (usually >50%) and overt PNH symptoms including clinical hemolysis, and a high risk of thrombosis [37]. In a study conducted over nearly 20 years involving 3085 patients with suspected myeloid disorders, 774 patients were positive for PNH clones and 327 of these had AA. In patients with AA and myelodysplastic syndrome (MDS), a high prevalence of small and very small PNH clones was found. The presence of clones (irrespective of size) was associated with a more favorable response to immunosuppressive therapy and overall survival [6], a finding reported in other studies and meta-analyses [33,38,39].

### 2.5. Paroxysmal Nocturnal Hemoglobinuria Screening and Guidance

PNH clones can emerge as a secondary event in patients with non-severe/moderate AA or may be present from the outset of the disease. In such cases, PNH clones represent an immunological challenge, constituting a potential adverse consequence across the full spectrum of bone marrow failure syndromes [20,40]. However, the risk of evolution into hemolytic PNH, as well as the possible predictive role for clinical outcomes, are not well defined in AA [41]. In a large series of patients with non-severe AA (*n* = 259) it was found that the majority required immunosuppressant therapy (with or without eltrombopag), and only one-third could be monitored with follow-up alone. Hemolytic PNH occurred in 10% of cases, being associated with the detection of small clones at diagnosis [40]. These findings highlight the need for effective screening programs, which may be more important in Asian countries as the prevalence of PNH appears to be higher [42]. This is also highlighted by the fact that accurate diagnosis may be key to an optimal treatment strategy. For example, in the absence of hemolysis or PNH symptoms, patients with aAA and smaller PNH clones do not benefit from complement inhibitors but should be monitored for any changes in their disease status [37]. In contrast, complement inhibitors improve outcomes in patients with larger PNH clones, who invariably have the classical disease associated with clinical hemolysis, and a high burden of PNH-associated symptoms.

In 2010, the International Clinical Cytometry Society recommended the use of multiparameter flow cytometry as it increases the sensitivity of detecting PNH cells by 100-fold [43]. As noted above, the detection and monitoring of PNH clones in patients with AA or MDS may play a pivotal role in determining treatment strategies. An individualized approach is required when evaluating patients, and referral to a specialized center with expertise in managing PNH and AA is recommended. Numerous guidelines/guidance papers have been published in relation to PNH screening in aAA patients and some key findings are summarized in Table 1 [6,7,8,9,10,11,12,13,14,15,16]. The guideline groups are consistent in recommending that screening for PNH clones be performed at the time of AA diagnosis. Some groups also recommend monitoring for PNH clones during follow-up of AA patients, although the suggested frequency of such monitoring varies (Table 2).

## 3. Monitoring Paroxysmal Nocturnal Hemoglobinuria Clones in Patients with Aplastic Anemia: How and When?

This section summarizes the main tests used to monitor PNH clones in patients with AA, followed by recommendations for monitoring specific to Asian countries.

### 3.1. General Considerations

Small PNH clones are found in up to 50% of patients with AA [26,29] and may remain stable or may increase or decrease in size over time [9]. It is important to identify any large, clinically significant PNH clones, especially if they are associated with evidence of hemolysis [9]. There is also evidence that the presence of a PNH clone predicts a higher response rate to immunosuppressive therapy [33,38,39]. In addition, the presence of a PNH clone at the time of AA diagnosis is associated with an increased risk for subsequent development of PNH/AA-PNH syndrome after immunosuppressive therapy, highlighting the need for more frequent flow cytometry testing in these patients [33].

Guidelines on the management of AA [9] and PNH [7,8,10,11,12,13,14,15,16] recommend screening for PNH clones at the time of an AA diagnosis (Table 1). Several guidelines also recommend flow cytometry testing for PNH during follow-up of patients with AA to monitor clone size in patients known to have clones and to screen for the development of PNH in other patients [7,8,9,11,14]. There is currently no consensus on the optimal frequency of monitoring for different patient subgroups. Available guidelines recommend intervals of 3–6 months for at least 2 years, increasing to 1-year intervals for patients who are stable (Table 2).

### 3.2. Flow Cytometry

The standard technique for assessing PNH clones is flow cytometry of peripheral blood samples. Increased sensitivity has been achieved using fluorescent aerolysin (FLAER)-based tests [44]. The diagnosis of PNH requires the demonstration of at least two different GPI-anchored protein (GPI-AP) deficiencies in two different cell lines (among granulocytes, monocytes and red blood cells) [8,10]. Evaluation of leukocytes (granulocytes and monocytes) produces more accurate estimates of PNH clone size compared with evaluation of red blood cells, as results for the latter can be affected by major hemolysis or red blood cell transfusions [7,16]. FLAER assays can be used in white blood cell studies and are often combined with monoclonal antibodies specific for GPI-AP that are normally expressed on granulocytes or monocytes [45].

Different immunophenotypic panels are available, with varying sensitivities for the detection of PNH cells. Low-sensitivity assays (that can detect cells at a frequency of 1%) are acceptable for the diagnosis of classical hemolytic PNH, in which moderate or large clones are typically present; however, high-sensitivity assays (that can detect cells at a frequency of 0.01%) are needed to detect small populations of PNH clones in patients with AA [10,45].

As flow cytometry testing incurs substantial costs, a balance must be struck between robust PNH testing and economic considerations [45]. Such testing may not be readily available or affordable in all settings [46,47]. In particular, the use of flow cytometry to monitor PNH clones during long-term follow-up carries cost implications with respect to performing the test and the need for a suitable diagnostic facility. This may be a barrier to its implementation in resource-limited countries in the Asia–Pacific region. The cost of flow cytometry testing also has implications as regards insurance coverage/reimbursement in some countries, e.g., Japan [16]. In order for the dissemination of flow cytometry to be a reality in many countries around the world, the availability of low-cost models is essential. The use of a FLAER-only panel has been suggested as a sensitive and cost-effective option for identifying PNH clones, especially for resource-limited settings [44,48]. Steps should also be taken to reduce other costs, such as those associated with identifying low-risk patients triaged on the basis of parameters such as age, complete blood cell count, etc. Successful triage requires trained staff to interpret the data and identify patients at the greatest risk [49].

### 3.3. Other Tests

Standard blood tests able to be performed regularly during follow-up of AA patients, and which can assist in identifying patients who may need flow cytometry testing, include LDH level and reticulocyte count, both of which are markers of hemolysis in PNH.

LDH is an enzyme located in the cytoplasm of cells, with two isoenzymes, LDH-1 and LDH-2, particularly expressed in red blood cells [50]. The level of LDH measured in serum reflects cellular turnover. A marked increase in serum LDH (up to 4–5 times the upper limit of normal [ULN]) is seen in intravascular hemolysis (compared with only a small increase in extravascular hemolysis) [50]. In PNH, an increase in LDH to ≥1.5 times ULN is generally seen in untreated patients [8,16]. Some studies indicate that Asian patients tend to have higher increases in LDH than patients in other world regions. Data from the international PNH registry indicated that the median LDH level in newly diagnosed PNH patients in Taiwan was 4.9-fold the ULN compared with 2-fold for patients from the rest of the world, and that 80% of Taiwanese patients had LDH >1.5-fold ULN versus 58.6% for the rest of the world [51]. Similar data were reported for a Korean registry, with a median increase in LDH of 4.1-fold ULN and an LDH level ≥1.5 × ULN present in 76.3% of patients [52].

Reticulocytes are non-nucleated precursors of red blood cells that provide an index of bone marrow hematopoietic activity [50]. The reticulocyte count reflects intravascular hemolysis, extravascular hemolysis and bone marrow reserve. The count usually increases in response to hemolysis, due to compensatory reticulocytosis. However, in patients with hemolytic conditions who also have a bone marrow failure syndrome, the compensatory increase in reticulocytes may be reduced or absent [50]. In patients with PNH, the absolute reticulocyte count is generally elevated [50,53]; however, this may not be seen in patients who have both AA and PNH [53].

LDH and reticulocyte counts are parameters that can be measured by most standard laboratories, either alone or as part of general biochemical and hematological test panels, at low cost. They are widely available in all countries, including those with limited resources. In addition, peripheral blood smears are often performed as part of the diagnostic work-up and evaluation of patients with PNH [45]. Reticulocytes can be identified using stains containing methylene blue. A manual reticulocyte count on a blood smear may be a useful test to raise suspicion of PNH and for ongoing monitoring in resource-limited environments.

### 3.4. Recommendations for Asia

The expert panel noted that flow cytometry is an expensive test, and their recommendations take this into account. In their experience, after screening for PNH using flow cytometry at the time of AA diagnosis, monitoring for PNH during follow-up initially involves measurement of LDH. This is especially the case for AA patients found to be negative for PNH clones; in this scenario, the panel members do not monitor frequently using flow cytometry because it is not common for PNH clones to expand from 0 to more than 10%.

The panel made the following recommendations. Screening for PNH should be performed at the time of an AA diagnosis. During follow-up, monitoring for PNH by flow cytometry should be performed at least once every few years (2–3 years) in AA patients in whom PNH-type blood cells were not detected in the initial analysis. LDH and reticulocytes should be measured at every visit. In the event of a high LDH level (with or without a high reticulocyte count) or the development of significant symptoms such as thrombosis, on-demand flow cytometry should be performed. If testing is negative, return to monitoring via clinical laboratory tests and symptoms. Once PNH-type blood cells are detected, subsequent periodic monitoring by flow cytometry should be performed at least once a year.

With respect to the definition of clone size, the panel noted that the cutoff for a clinically meaningful PNH clone size varies between countries, often depending on national reimbursement requirements. For example, the cut-off in red blood cells is 1% in Japan, but 10% in many countries. The proposed tests and frequency of monitoring for PNH in patients with AA are summarized in Table 3. An algorithm summarizing key steps is provided in Figure 1. In addition, a D-dimer test to exclude active thrombosis may help detect subclinical clots in patients with elevated serum LDH.

For resource-limited settings, the panel recommended monitoring the reticulocyte count and LDH level at every visit as the minimum, followed by flow cytometry if there is evidence of hemolytic PNH. In some countries, reimbursement may not be available for flow cytometric analysis of PNH-type cells in both granulocytes and red blood cells. D-dimer testing may be considered in patients with elevated LDH.

## 4. Treatment of Paroxysmal Nocturnal Hemoglobinuria in Aplastic Anemia Patients

This section summarizes the main treatments for AA and PNH as separate disorders, followed by the treatment of patients with both PNH and AA, and ends with treatment recommendations for Asian countries.

### 4.1. Treatment of Aplastic Anemia

Allogeneic hematopoietic stem-cell transplant (HSCT) is a potentially curative treatment for patients with severe AA, although it may not be appropriate for older adults and patients with significant comorbidities, as well as those with non-severe AA [9,54]. The first-line immunosuppressive therapy for older patients with severe AA and younger patients who lack a matched donor is a combination of anti-thymocyte globulin plus cyclosporine [9,55]. The addition of eltrombopag, a thrombopoietin receptor agonist, to standard immunosuppressive therapy can improve the response in patients with severe AA [56,57]. Studies in Asian AA populations indicate outcomes after HSCT or immunosuppressive therapy similar to those reported for Western populations [58,59]. Geographical region (Europe/Asia) was not a significant predictor of survival after immunosuppressive therapy [60].

### 4.2. Treatment of Paroxysmal Nocturnal Hemoglobinuria

Complement inhibitors (e.g., eculizumab, ravulizumab) are the main treatment for classical PNH [12,16]. Treatment is generally indicated for symptomatic patients with hemolysis and evidence of high disease activity [10,12,15]. Eculizumab and ravulizumab have been shown to be effective in Asian patients in clinical trials and observational studies [5,61,62,63,64,65,66], with efficacy and safety outcomes consistent with those reported for global populations in clinical trials [5]. Although it can be curative, HSCT is not a first-line therapy for PNH due to the risk of treatment-related morbidity and mortality [10,12,15].

Patients with subclinical PNH do not need specific PNH treatment. For patients with PNH in the context of another bone marrow disorder, the initial focus is usually on the bone marrow disorder. Patients with a large PNH clone may benefit from eculizumab; see below for additional information.

### 4.3. Treatment of Patients with Paroxysmal Nocturnal Hemoglobinuria and Aplastic Anemia

Few studies have focused on treatment options for patients with both PNH and AA, and very few studies involved Asian patients. Clinical studies in Japan and Korea found that eculizumab showed similar efficacy in patients with PNH plus bone marrow dysfunction (AA or MDS) and patients with PNH alone [61,62,67]. International and European observational studies found that the concomitant administration of eculizumab and immunosuppressive therapy in patients with concurrent PNH and AA had no adverse impact on the effectiveness of treatment, regardless of the sequence of use, and did not raise any safety concerns [68,69,70,71]. Studies of allogeneic HSCT in patients with PNH in China and in Turkey found that rates of overall survival and graft-failure-free survival did not differ significantly between patients with classical PNH and those with PNH-AA syndrome [72,73]. A US study found that eculizumab administered as a bridge to HSCT in patients with severe AA and PNH was safe and efficacious [74].

Available guidelines for the management of PNH or AA recommend that, for patients with PNH in the context of a bone marrow failure disorder such as AA, the focus of treatment should generally be on the underlying bone marrow disorder [10,15]. The presence of a small/moderate PNH clone should not directly affect the choice of treatment for the bone marrow failure, including allogeneic HSCT or immunosuppressive therapy [9]. However, patients with large clones may benefit from eculizumab [15]; hence, both complement inhibition and immunosuppressive therapy may be needed in some patients [12]. Complement inhibition is not required for patients with AA and small clones, but these patients should be monitored for the development of symptomatic PNH [12]. If PNH clonal expansion occurs in a patient with AA, leading to hemolysis or thrombosis, the patient should be regarded as having classical PNH and treated with a complement inhibitor [10,16]. HSCT may be considered in patients with severe AA who have a PNH clone [10,12,16].

### 4.4. Recommendations for Asia

Criteria that may influence the treatment plan for patients with both PNH and AA include clone size, the severity of anemia, the severity of other cytopenias related to bone marrow failure, the severity of hemolysis and the presence of thrombosis.

Members of the panel highlighted points about current practices in various countries. Japan has guidelines for both PNH and AA; if bone marrow failure is dominant in a PNH patient, the AA guidelines are followed. Bone marrow transplantation is limited to young patients who have a disease condition affecting their life prognosis, such as severe AA or recurrent thrombosis [16], although the latter setting requires careful consideration in view of evidence associating a history of thromboembolism with decreased overall survival post-transplantation [75]. Singapore follows the UK guidelines for AA. Transplantation is an option for patients with severe AA, provided the patient is under 40 years of age. In Korea, patients who receive a complement 5 inhibitor (eculizumab or ravulizumab) must be treated at a hospital where they can undergo transplantation. Decisions about transplantation or treatment with a complement inhibitor can be made only at that hospital and not at any other hospital. As local guidelines on anticoagulant therapy in PNH patients are not available in Taiwan, international guidance is followed.

The panel made several consensus recommendations for treating patients with both PNH and AA (summarized in Table 4). Measuring PNH clones in patients with AA can assist with the choice of initial AA treatment. A positive PNH clone may indicate an underlying immunological background for the patient, and immunosuppressive therapy may be desirable in this setting. The treatment of patients with both PNH and AA depends on which condition predominates, hemolysis or bone marrow failure. If hemolysis is predominant, then PNH is the predominant disorder, and treatment guidelines for PNH should be followed. If a PNH clone is present but the degree of bone marrow failure is severe, then regardless of whether hemolysis is mild or moderate, guidelines for treating AA should be followed. HSCT is an option for patients with severe AA in the absence of any other treatment options. The decision should be made after explanation and discussion with the patient and family. HSCT carries the risk of transplant-related mortality or morbidity; hence, it is generally reserved for cases in which there are no other treatment plans for a patient.

## 5. Challenges and Opportunities

Given the high incidence of PNH in patients with AA, and the knowledge that PNH may exacerbate morbidity and mortality, early diagnosis is important. Moreover, PNH disease status may have an impact on the overall treatment plan for the patient. Most current guidelines for the management of PNH or AA provide only limited guidance on the monitoring of PNH clones and clone size in patients with AA. The panel of experts from Asia noted that available guidance may not be practical for healthcare systems in the Asia–Pacific region. In addition, there is limited information on the best treatment strategy for patients who have both PNH and AA, including prioritization, sequencing and/or concomitant use of treatments for the disorders.

The panel agreed that it would be beneficial to have recommendations specific to the Asia–Pacific region in order to take account of local healthcare systems. It was noted that the availability of such guidelines may assist with healthcare insurance reimbursement for PNH/AA patients, especially in resource-limited settings. A particular challenge in lower-income countries is limited or no access to anticomplement therapy, which makes the detection of PNH less useful clinically. In some countries, such as Thailand, pharmacoeconomic analysis and price negotiation processes are usually required for reimbursement considerations.

In making their recommendations for the Asia–Pacific region, the expert panel considered available guidelines on PNH and AA (only one of which is from an Asia–Pacific country, Japan), as well as healthcare systems and reimbursement requirements for countries in the region. They also considered published evidence for Asian patients with AA and PNH, although few such data were available. It is apparent that additional studies are needed on the clinical management of PNH in AA patients among Asian populations, as is research into the health economic aspects of monitoring and treating this patient population in countries within the Asia–Pacific region. Data from national registries may also provide additional information on patients with PNH/AA in the region.

The expert panel’s discussion and recommendations focused specifically on PNH in patients with AA. However, the recommendations could be extended to patients with other bone marrow failure disorders, such as MDS, a proportion of whom exhibit PNH clones [6,76]. Patients with lower-risk MDS, especially those with marrow hypocellularity, may undergo similar treatment to patients with AA [77,78]. Guidelines recommend screening for PNH in MDS patients who have evidence of hemolysis, hypoplastic bone marrow or refractory cytopenia [8,10].

In light of the diverse PNH monitoring recommendations for AA patients in current guidelines, moving forward, experts in the field from across the world should attempt to align the guidance on monitoring and treating PNH in patients with AA. There are currently insufficient data from well-designed clinical trials to support the development of a single guideline based on evidence levels. However, it is anticipated that experts with experience in evaluating and managing patients with AA and PNH might be able to arrive at an international consensus on best practice for patient care, based on a review of the available evidence, existing guidance documents, and their clinical experience. In this context, it would be important that consideration be given to the feasibility of implementing recommendations in different healthcare and reimbursement systems, including in resource-limited countries.

## 6. Conclusions

Available guidance on the monitoring strategy for PNH in AA patients is limited, inconsistent in some aspects and may not be practical for all healthcare systems. In addition, there is limited information regarding the overall treatment strategy for patients with both PNH and AA. An expert panel has provided recommendations on the monitoring and treatment of PNH in patients with AA that are practical for healthcare systems in the Asia–Pacific region.

## Figures and Tables

**Figure 1 ijms-25-12160-f001:**
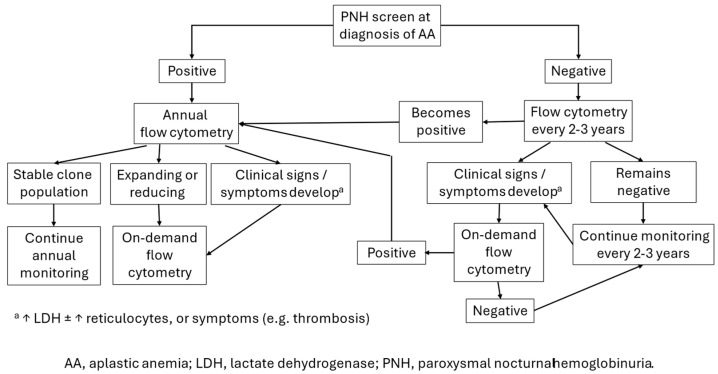
Algorithm for paroxysmal nocturnal hemoglobinuria screening and monitoring in patients with aplastic anemia in Asia.

**Table 1 ijms-25-12160-t001:** Overview of guidance regarding screening for/monitoring of paroxysmal nocturnal hemoglobinuria in patients with aplastic anemia.

**IPIG** ^a^ [7]	**ICCS** ^a^ [11]	**Canada** ^b^ [12]	**Delphi Panel** ^b^ [13]
At diagnosis of AA, flow cytometric analysis of erythrocytes and granulocytes is recommended After establishment of the diagnosis, flow cytometric analysis is recommended every 6 months for 2 years and yearly thereafter if parameters are stable If there is evidence of clinical progression (or amelioration), more immediate analysis should be performed Every year, even in the absence of clinical evidence of hemolysis (including patients treated with immunosuppressive therapy)	Patients with unexplained cytopenia in whom AA is a differential diagnostic consideration should be tested for PNH at the time of diagnosis Patients with no detectable clone should be screened every 6 months, decreasing to yearly if no clone appears in the first two years If a clone is present or appears, patients should be screened every 3 months until the clone size is shown to be stable for 2 years	All patients with a diagnosis or suspicion of AA: Testing should be done at diagnosis and monitored at least every 6 months	At initial diagnosis
**Germany** ^a^ [14]	**Belgium** ^a^ [10]	**Brazil** ^b^ [15]	**Turkey** ^a^ [8]
Diagnosis or strong suspicion of AA If there is evidence of a significant GPI-deficient cell population, the analysis should be repeated at intervals of six months, and then annually if the course is stable Every 12 months, provided there is no evidence of significant hemolysis	Patients with AA	Suspected or confirmed diagnosis of AA	PNH screening for all patients with AA is necessary even without hemolysis Every 6 months for patients with positive results, whereas annual follow-up is recommended for patients with negative results
**UK** ^a,c^ [9]	**Japan** ^a^ [16]		
Patients should be screened for PNH at the diagnosis of AA If persistently negative, test 6-monthly for 2 years then move to annual testing unless symptoms/signs develop If the PNH screen is, or becomes, positive, test 3-monthly for the first 2 years and reduce the frequency only if the proportion of PNH cells has remained stable	Patients with AA A finding of hemoglobinuria, obvious unexplained thrombosis, AA, low-risk MDS, or bone marrow failure with any cytopenia requires flow cytometry testing		

^a^ Guideline. ^b^ Consensus statement. ^c^ Guideline on AA. All other guidance documents are on PNH. AA, aplastic anemia; GPI, glycosylphosphatidylinositol; ICCS, International Clinical Cytometry Society; IPIG, International PNH Interest Group; MDS, myelodysplastic syndrome; PNH, paroxysmal nocturnal hemoglobinuria.

**Table 2 ijms-25-12160-t002:** Summary of guidance on the frequency of paroxysmal nocturnal hemoglobinuria clone monitoring in patients with aplastic anemia.

Frequency of Clone Monitoring		IPIG ^a^ [7]	ICCS ^a^ [11]	Canada ^b^ [12]	Delphi Panel ^b^ [13]	Germany ^a^ [14]	Belgium ^a^ [10]	Brazil ^b^ [15]	Turkey ^a^ [8]	UK ^a,c^ [9]	Japan ^a^ [16]
**At AA diagnosis**		✓	✓	✓	✓	✓	✓	✓	✓	✓	✓
**During follow-up**	Positive for clone	q6m	q3m	q6m		q6m				q3m	
	Negative for clone		q6m							q6m	
	Duration of monitoring	2 years	2 years							2 years	
	Stable clone	Annually	Annually			Annually				Annually	

^a^ Guideline. ^b^ Consensus statement. ^c^ Guideline on AA. All other guidance documents are on PNH. AA, aplastic anemia; ICCS, International Clinical Cytometry Society; IPIG, International PNH Interest Group; PNH, paroxysmal nocturnal hemoglobinuria; q3m, every 3 months; q6m, every 6 months.

**Table 3 ijms-25-12160-t003:** Frequency and modality of monitoring for paroxysmal nocturnal hemoglobinuria during follow-up of patients with aplastic anemia ^a^ in Asia.

Monitoring Modality	Proposed Frequency of Monitoring
Symptom evaluation	Every visit
Plasma/serum LDH level	With every blood test
Reticulocyte count	With every blood test
PNH clone size by flow cytometry	Periodic testing (at least once every 2–3 years) In the event of a rise in LDH level and/or reticulocyte count, or development of significant symptoms such as thrombosis, perform on-demand flow cytometry. (If negative, return to monitoring using clinical laboratory tests/symptoms) Once PNH-type blood cells are detected, annual flow cytometry testing is sufficient, provided there are no changes in clinical laboratory tests/symptoms

^a^ Patients in whom PNH-type blood cells were not detected in the initial screening at the time of diagnosis of AA. AA, aplastic anemia; LDH, lactate dehydrogenase; PNH, paroxysmal nocturnal hemoglobinuria.

**Table 4 ijms-25-12160-t004:** Treatment recommendations for patients with both paroxysmal nocturnal hemoglobinuria and aplastic anemia in Asia.

AA Patients Who Are Positive for PNH-Type Cells	Immunosuppressive Therapy Is Desirable
AA + PNH patients in whom hemolysis (PNH) is predominant	Follow published treatment guidelines for PNH
AA patients with PNH-type cells who have severe bone marrow failure (regardless of whether hemolysis is mild or moderate)	Follow published treatment guidelines for AA
Patients with severe AA for whom no other treatment options are available	Consider hematopoietic stem-cell transplant

AA, aplastic anemia; PNH, paroxysmal nocturnal hemoglobinuria.

## Data Availability

This guidance review and all the references supporting it will be uploaded to NCBI (PubMed).

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
