# Peer review of "Monitoring and Treatment of Paroxysmal Nocturnal Hemoglobinuria in Patients with Aplastic Anemia in Asia: An Expert Consensus"

_ijms, 2024, doi:10.3390/ijms252212160_

Round 1

Reviewer 1 Report

Comments and Suggestions for Authors

This is a very interesting paper on a subject of particular interest for Asian clinicians and patients, i.e. monitoring and treatment of paroxysmal nocturnal hemoglobinuria in patients with aplastic anemia in Asia. However, some issues need to be declared by the authors. This paper is entitled as expert consensus and also the authors give recommendations and algorithms. It is important to include a material and methods section where the authors should give in detail the way the paper was constructed, by including (1) the composition of the panel, (2) the way consensus was reached (voting anonymous or not, how many meetings took place etc), (3) how were the questions generated, (4) how was the literature searched (systematically vs narratively), (5) how was the quality of evidence evaluated (eg GRADE) etc.

Author Response

Author's Reply to the Review Report (Reviewer 1)

Comments and Suggestions for Authors

This is a very interesting paper on a subject of particular interest for Asian clinicians and patients, i.e. monitoring and treatment of paroxysmal nocturnal hemoglobinuria in patients with aplastic anemia in Asia. However, some issues need to be declared by the authors. This paper is entitled as expert consensus and also the authors give recommendations and algorithms.

It is important to include a material and methods section where the authors should give in detail the way the paper was constructed, by including (1) the composition of the panel, (2) the way consensus was reached (voting anonymous or not, how many meetings took place etc), (3) how were the questions generated, (4) how was the literature searched (systematically vs narratively), (5) how was the quality of evidence evaluated (eg GRADE) etc.

Submission Date: 21 October 2024

Date of this review: 27 Oct 2024 18:02:17

RESPONSE: Thank you for your comments. The following text has been added to Section 1 Introduction:

“Panel members experienced in the treatment of patients with PNH and AA were drawn from departments of Medicine, Hematology and Hematology–Oncology. In preparation for the meeting, a search of PubMed was conducted to identify relevant literature on the monitoring and treatment of patients with PNH and AA, including national/international guidance. During the meeting, the panel reviewed the available literature, discussed current practice in their countries, and addressed the following questions: (a) How do you screen and monitor for PNH among AA patients? What indicators should be monitored and how frequently? Should clone size be monitored regularly? What cutoff should be used for a clinically meaningful PNH clone size?; (b) What criteria drive AA treatment plan changes in patients with PNH? What should be the overall treatment and monitoring plan for patients with PNH and AA? Which guidelines, if any, do you follow in your practice? Based on the outcome of the discussion, the panel made recommendations for PNH monitoring and management in Asian patients with AA”.

Author's Reply to the Review Report (Reviewer 2)

Comments and Suggestions for Authors

This is a well written and interesting work on the follow up and the management of aplastic anemia in Asia. The authors present all the necessary data about the methods for the aplastic anemia follow up, especially regarding the involvement of PNH in Asia, the treatment options and the challenges for the local health systems. The authors have organized very well their presentation, so that the reader has all the necessary information.

Submission Date: 21 October 2024

Date of this review: 26 Oct 2024 11:21:08

RESPONSE: Thank you for your comments.

Author's Reply to the Review Report (Reviewer 3)

Comments and Suggestions for Authors

Thank you for the opportunity to review this article.

Minor suggestions:

Introduction: please add a paragraph on the prevalence of both of the diseases mentioned in the review; it helps highlight the importance of your work.

RESPONSE: Text has been added at the start of Section 1 Introduction and the original first sentence has been amended:

“Paroxysmal nocturnal hemoglobinuria (PNH) and aplastic anemia (AA) are rare disorders, with reported incidences of 1–3.5 cases/million people/year [1,2] and 2–7.4 cases/million/year [3], respectively. Both diseases are more common in Asian than in Western populations [1,3]. PNH clones can be identified in a significant proportion of patients with AA [4–6]”.

Background: well written, nothing much to add

Table 1. Well researched, nothing much to add

General considerations: consider talking about the role of the blood smears and the methylene blue wet mount for reticulocytes as a primary test to raise suspicion.

RESPONSE: Text has been added to Section 3.3 Other Tests:

“In addition, peripheral blood smears are often performed as part of the diagnostic work-up and evaluation of patients with PNH [45]. Reticulocytes can be identified using stains containing methylene blue. A manual reticulocyte count on a blood smear may be a useful test to raise suspicion of PNH and for ongoing monitoring in resource-limited environments.”

Table 2. for me it's a little hard to read, insert somewhere in the top left corner "Frequency of clone monitoring". Due to the fact it starts with "at aa diagnosis" it's quite confusing, especially if you don't read the table header.

RESPONSE: Text has been added to the top left corner of Table 2: "Frequency of clone monitoring".

References are up to date, well chosen.

One last minor suggestion is to include a chapter on the role of the peripheral blood smear as well and the importance of doing it with each flow cytometry examination. Maybe even consider adding some pictures of blood smears from patients with AA and PNH. Not everyone has a flow cytometer available (sadly) but everyone in the world can perform a blood smear and do a quick manual reticulocyte count.

RESPONSE: Text has been added to Section 3.3 Other Tests:

“In addition, peripheral blood smears are often performed as part of the diagnostic work-up and evaluation of patients with PNH [45]. Reticulocytes can be identified using stains containing methylene blue. A manual reticulocyte count on a blood smear may be a useful test to raise suspicion of PNH and for ongoing monitoring in resource-limited environments.”

Besides these minor suggestions, there's nothing much i can add. The article is really well researched and written and i think it brings all the relevant information into one article.

Due to this reason, i recommend accepting this article with minor changes.

RESPONSE: Thank you for your comments.

Wishing you the best,

Reviewer :)

Submission Date: 21 October 2024

Date of this review: 27 Oct 2024 14:59:33

Reviewer 2 Report

Comments and Suggestions for Authors

This is a well written and interesting work on the follow up and the management of aplastic anemia in Asia. The authors present all the necessary data about the methods for the aplastic anemia follow up, especially regarding the involvement of PNH in Asia, the treatment options and the challenges for the local health systems. The authors have organized very well their presentation, so that the reader has all the necessary information.

Author Response

Author's Reply to the Review Report (Reviewer 2)

Comments and Suggestions for Authors

This is a well written and interesting work on the follow up and the management of aplastic anemia in Asia. The authors present all the necessary data about the methods for the aplastic anemia follow up, especially regarding the involvement of PNH in Asia, the treatment options and the challenges for the local health systems. The authors have organized very well their presentation, so that the reader has all the necessary information.

Submission Date: 21 October 2024

Date of this review: 26 Oct 2024 11:21:08

RESPONSE: Thank you for your comments.

Reviewer 3 Report

Comments and Suggestions for Authors

Thank you for the opportunity to review this article.

Minor suggestions:

Introduction: please add a paragraph on the prevalence of both of the diseases mentioned in the review; it helps highlight the importance of your work. 

Background: well written, nothing much to add

Table 1. Well researched, nothing much to add

General considerations: consider talking about the role of the blood smears and the methylene blue wet mount for reticulocytes as a primary test to raise suspicion. 

Table 2. for me it's a little hard to read, insert somewhere in the top left corner "Frequency of clone monitoring". Due to the fact it starts with "at aa diagnosis" it's quite confusing, especially if you don't read the table header. 

References are up to date, well chosen. 

One last minor suggestion is to include a chapter on the role of the peripheral blood smear as well and the importance of doing it with each flow cytometry examination. Maybe even consider adding some pictures of blood smears from patients with AA and PNH. Not everyone has a flow cytometer available (sadly) but everyone in the world can perform a blood smear and do a quick manual reticulocyte count. 

Besides these minor suggestions, there's nothing much i can add. The article is really well researched and written and i think it brings all the relevant information into one article. Due to this reason, i recommend accepting this article with minor changes.

Wishing you the best,

Reviewer :)

Author Response

Author's Reply to the Review Report (Reviewer 3)

Comments and Suggestions for Authors

Thank you for the opportunity to review this article.

Minor suggestions:

Introduction: please add a paragraph on the prevalence of both of the diseases mentioned in the review; it helps highlight the importance of your work.

RESPONSE: Text has been added at the start of Section 1 Introduction and the original first sentence has been amended:

“Paroxysmal nocturnal hemoglobinuria (PNH) and aplastic anemia (AA) are rare disorders, with reported incidences of 1–3.5 cases/million people/year [1,2] and 2–7.4 cases/million/year [3], respectively. Both diseases are more common in Asian than in Western populations [1,3]. PNH clones can be identified in a significant proportion of patients with AA [4–6]”.

Background: well written, nothing much to add

Table 1. Well researched, nothing much to add

General considerations: consider talking about the role of the blood smears and the methylene blue wet mount for reticulocytes as a primary test to raise suspicion.

RESPONSE: Text has been added to Section 3.3 Other Tests:

“In addition, peripheral blood smears are often performed as part of the diagnostic work-up and evaluation of patients with PNH [45]. Reticulocytes can be identified using stains containing methylene blue. A manual reticulocyte count on a blood smear may be a useful test to raise suspicion of PNH and for ongoing monitoring in resource-limited environments.”

Table 2. for me it's a little hard to read, insert somewhere in the top left corner "Frequency of clone monitoring". Due to the fact it starts with "at aa diagnosis" it's quite confusing, especially if you don't read the table header.

RESPONSE: Text has been added to the top left corner of Table 2: "Frequency of clone monitoring".

References are up to date, well chosen.

One last minor suggestion is to include a chapter on the role of the peripheral blood smear as well and the importance of doing it with each flow cytometry examination. Maybe even consider adding some pictures of blood smears from patients with AA and PNH. Not everyone has a flow cytometer available (sadly) but everyone in the world can perform a blood smear and do a quick manual reticulocyte count.

RESPONSE: Text has been added to Section 3.3 Other Tests:

“In addition, peripheral blood smears are often performed as part of the diagnostic work-up and evaluation of patients with PNH [45]. Reticulocytes can be identified using stains containing methylene blue. A manual reticulocyte count on a blood smear may be a useful test to raise suspicion of PNH and for ongoing monitoring in resource-limited environments.”

Besides these minor suggestions, there's nothing much i can add. The article is really well researched and written and i think it brings all the relevant information into one article.

Due to this reason, i recommend accepting this article with minor changes.

Round 2

Reviewer 1 Report

Comments and Suggestions for Authors

The authors have adequately replied to the reviewers' comments.